# HYDRA: Hybrid Robot Actions
# for Imitation Learning

**Suneel Belkhale, Yuchen Cui, Dorsa Sadigh**
Stanford University, Stanford CA
{belkhale, yuchenc, dorsa}@stanford.edu

**Abstract:** Imitation Learning (IL) is a sample efficient paradigm for robot learning using expert demonstrations. However, policies learned through IL suffer from state distribution shift at test time, due to compounding errors in action prediction which lead to previously unseen states. Choosing an action representation for the policy that minimizes this distribution shift is critical in imitation learning. Prior work propose using temporal action abstractions to reduce compounding errors, but they often sacrifice policy dexterity or require domain-specific knowledge. To address these trade-offs, we introduce HYDRA, a method that leverages a hybrid action space with two levels of action abstractions: *sparse high-level waypoints* and *dense low-level actions*. HYDRA dynamically switches between action abstractions at test time to enable both coarse and fine-grained control of a robot. In addition, HYDRA employs action relabeling to increase the consistency of actions in the dataset, further reducing distribution shift. HYDRA outperforms prior imitation learning methods by 30-40% on seven challenging simulation and real world environments, involving long-horizon tasks in the real world like making coffee and toasting bread. Videos are found on our website: https://tinyurl.com/3mc6793z

## 1 Introduction

In recent years, supervised learning methods have made remarkable advancements in computer vision (CV), natural language processing (NLP), and human-level game playing [1, 2, 3, 4, 5, 6, 7]. In robotics, *imitation learning* (IL) has emerged as a data-driven and sample efficient approach for programming robots using expert demonstrations. More specifically, behavioral cloning (BC) methods treat IL as a supervised learning problem and directly train a policy to map states to actions. BC methods are often favored in practice for their simplicity but suffer from the well-known distribution shift problem, where the test time state distribution deviates from the training state distribution, primarily caused by the accumulation of errors in action predictions [8, 9, 10].

Broadly, prior work has explored reducing distribution shift by interactively adding new data [9], incorporating large prior datasets [11, 12], choosing better state representations (inputs) [13, 14], or altering model or loss structure [15, 16, 14]. A less explored but critical factor is the *action* representation (outputs): action prediction error partially stems from how difficult it is for the policy to capture the expert demonstrated actions, so action representations are a critical line of defense against distribution shift. Prior work studying action representations generally fall into two camps: (1) methods that use *temporal abstractions* to treat long action sequences as a single action (i.e., reducing the effective task horizon) and thus reduce the potential for compounding errors, and (2) methods that make the action representation more *expressive* to minimize the single-step prediction error [17, 18, 16, 19, 15]. However, both approaches come with a number of shortcomings.

Methods using *temporal abstractions* often come at the cost of either the dexterity of the robot or the generality to new settings. One prior approach is for the robot to follow waypoints that cover multiple time steps [17, 14]; however, waypoints alone are not reactive enough for dynamic, dexterous action sequences (e.g., inserting a coffee pod). Other works use structured movement primitives that can capture more dynamic behaviors like skewering food items or helping a person

7th Conference on Robot Learning (CoRL 2023), Atlanta, USA.

get dressed [20, 18, 21], but relying on pre-defined primitives often sacrifice generalizability to new settings (e.g., new primitives beyond skewering for food manipulation). Today, we lack temporal abstractions that reduce distribution shift without losing policy dexterity and generality.

Other methods design each action to be more *expressive* to capture the multi-modality present in human behavior [19, 15, 16]; however, these expressive action spaces often lead to overfitting, high training time, or complex learning objectives. Rather than making the policy more expressive, a more robust approach is to make the actions in the dataset more *consistent* at a given state and easier to learn (e.g., showing one consistent way to insert a coffee pod rather than many conflicting ways). Prior work shows that more action consistency (e.g., consistent human demonstrators) with sufficient state coverage lead to better policies [19, 14, 22], likely by reducing online policy errors [23].

To enable both a better temporal abstraction and more consistent actions in the dataset, our key insight is to leverage the fact that most robotics tasks are hierarchical in nature – they can be divided into two distinct *modes* of behaviors: *reaching high-level waypoints* (e.g., free-space motion) or *fine-grained manipulation* (e.g., object interaction). Then, we can learn a policy that dynamically switches between these modes – this is in fact similar to models of human decision making, where it is widely believed that humans can discover action abstractions and switch between them during a task [24, 25]. Capturing both waypoints and fine-grained actions enables us to compress action sequences (i.e., reduce distribution shift) without sacrificing the dynamic parts of the task, thus maintaining dexterity. In practice, this abstraction is general enough to represent most tasks in robot manipulation. Another notable advantage of partitioning tasks into two modes is that, during the waypoint reaching phase, we can *relabel* our actions with more consistent waypoint-following behaviors, thus increasing action consistency in the dataset.

Leveraging this insight, we propose HYDRA, a method that dynamically switches between two action representations: *sparse* waypoint actions for free-space linear motions and *dense*, single-step delta actions for contact-rich manipulation. HYDRA learns to switch between these action modes with human-labeled modes, which are provided after or during data collection with minimal additional effort. In addition, HYDRA *relabels* low-level actions in the dataset during the waypoint periods – where the robot is moving in free space (e.g., when reaching a coffee pod) to follow consistent paths. These consistent actions simplify policy learning, which reduces action prediction error in the dataset overall and thus reduces distribution shift. HYDRA outperforms baseline imitation learning approaches across seven challenging, long-horizon manipulation tasks spanning both simulation and the real world. In addition, it is able to perform a complex coffee making task involving many high precision stages with 80% success, 4x the performance of the best baseline, BC-RNN.

## 2   Related Work

**Data Curation**: Several prior works aim to *curate* data based on some notion of data quality, in order to reduce distribution shift. Most works define quality as the state diversity present in a dataset, To increase state diversity, Ross et al. [9] proposed to interactively collect *on-policy* demonstration data, but this requires experts to label actions for newly visited states. To reduce expert supervision, some methods use interventions to relabel on-policy data, where interventions can be automatically or human generated [26, 27, 28, 29, 22, 30]. Laskey et al. [31] inject noise during data collection to increase state diversity to achieve similar performance as interactive methods. Recent work has sought to formalize a broader notion of data quality beyond just state diversity [23]. HYDRA takes this broader definition into account, increasing data quality through action consistency.

**Model and State Priors**: Rather than changing the data, many prior works build in structure to the model itself to address distribution shift. Object-centric state representations have been shown to make policies more generalizable [13]. Similarly, pretrained state representations trained on multi-task data have been shown to improve sample efficiency and robustness [12, 32]. Adding structure into the model itself, for example using implicit representations or diffusion-based policies, has also been shown to improve performance [16, 15]. The changes in HYDRA affect the action space and thus are compatible with many of these prior approaches.

**Action Representations**: Another approach is to change the action representation to reduce compounding errors. One category of prior works leverage *temporal action abstractions* to reduce the

number of policy steps. Several works have learned skills from demonstrations, usually requiring lots of data but struggling to generalize [33, 34, 14]. Others use parameterized action primitives or motion primitives, but despite being more sample efficient, these often require privileged state information or are not general enough for all scenes [20, 18, 21]. Waypoint action spaces have also been shown to be a sample efficient temporal abstraction; however, they fail to capture dynamic and dexterous tasks in the environment [35, 16, 36]. Sparse waypoint labeling has also been shown to be easier for people than action labeling [36]. For more dexterity, Johns [37] proposes Coarse-to-Fine Imitation Learning by modeling a single demonstrated trajectory as two parts: an approaching trajectory followed by an interaction trajectory. Di Palo and Johns [38] extend this to allow multi-stage tasks, but this approach still relies on fully known and fixed "stages" of the task along with labeled bottleneck states. Also, due to the use of self-replay during interactions, these approaches would struggle with scene variation or generalization. HYDRA builds on these works, combining waypoints and low-level actions into one fully learned model to reduce compounding errors without losing dexterity or generality. Another category of works seek to increase the *expressivity* of a single action to reduce action prediction error, for example with Gaussian mixture models or energy models [19, 15, 16]. However, increasing expressivity often leads to overfitting, more complex learning objectives, and increased training and evaluation time. Instead of increasing expressivity, HYDRA takes a more robust approach by increasing action *consistency* in the data. Prior work shows the importance of consistent actions for minimizing distribution shift [19, 23]. HYDRA relabels actions in the dataset after data collection to increase consistency.

## 3    Preliminaries

Imitation learning (IL) assumes access to a dataset $\mathcal{D} = \{\tau_1, \ldots, \tau_n\}$ of $n$ expert demonstrations. Each demonstration $\tau_i$ is a sequence of observation-action pairs of length $N_i$, $\tau_i = \{(o_1, a_1), \ldots, (o_{N_i}, a_{N_i})\}$, with observations $o \in \mathcal{O}$ and actions $a \in \mathcal{A}$. $\mathcal{O}$ often consists of robot proprioceptive data such as end effector poses and gripper widths, denoted $s_p \in \mathcal{P}$, as well as environment observations such as images or object poses, denoted $s_e \in \mathcal{E}$, such that $\mathcal{O} = \mathcal{P} \oplus \mathcal{E}$. The true state of the environment is $s \in \mathcal{S}$. In robotics, the action space usually consists of either torque, velocity, or position commands for the robot. While velocity actions are most common, prior works also use position actions in the form of target waypoints [14, 35]. The IL objective is to learn a policy $\pi_\theta : \mathcal{O} \to \mathcal{A}$ mapping from observations to actions via the supervised loss:

$$\mathcal{L}(\theta) = -\mathbb{E}_{(o,a) \sim p_\mathcal{D}} \left[ \log \pi_\theta(a|o) \right] \tag{1}$$

At test time, the learned policy $\pi_\theta$ is rolled out under environment dynamics $f : \mathcal{S} \times \mathcal{A} \to \mathcal{S}$. Per step, we observe $o_t$, sample an action $\tilde{a}_t \sim \pi(\cdot|o_t)$, and obtain the next state $s_{t+1} = f(s_t, \tilde{a}_t)$.

**Distribution Shift in IL.** A fundamental challenge with imitation learning is state *distribution shift* between training and test time. Considering training sample $(\ldots o_t, a_t, o_{t+1} \ldots)$: if the learned policy outputs $\tilde{a}_t \sim \pi(\cdot|o_t)$, which has a small action error $\epsilon_t = \tilde{a}_t - a_t$, the next state following this action will also deviate: $\tilde{s}_{t+1} = f(s_t, a_t + \epsilon_t)$, which in turn affects the policy output at the next step. For real world dynamics, this change in next state can be highly disproportionate to $||\epsilon_t||$. For example in the coffee task in Fig. 1, with a slight change in gripper position (small $\epsilon_t$) the policy can misgrasp the coffee pod (large change in $s_{t+1}$ and $o_{t+1}$). Furthermore, noise in the dynamics $f$ can lead to even larger changes in $o_{t+1}$. As we continue to execute for the next $N - t$ steps, this divergence from the training distribution can compound, often leading to task failure.

Therefore, reducing distribution shift requires reducing $\epsilon_t$ for all $t \in \{1, \ldots, N\}$ or increasing the coverage of states $s_t$. One approach to reduce policy error is increasing **action consistency**, which prior work defines as lowering the entropy of the expert policy $\pi_E$ at each state: $\mathcal{H}_{\pi_E}(a|s)$ [23]. However, there is a trade-off between state coverage and action consistency during data collection, since less consistent actions often lead to more diverse states [23, 19]. HYDRA reduces distribution shift by using a temporal abstraction for the action space – which shortens the number of policy steps $N$ and thus reduces compounding errors – and by improving action consistency in offline data – which reduces $||\epsilon_t||$ without reducing state coverage.

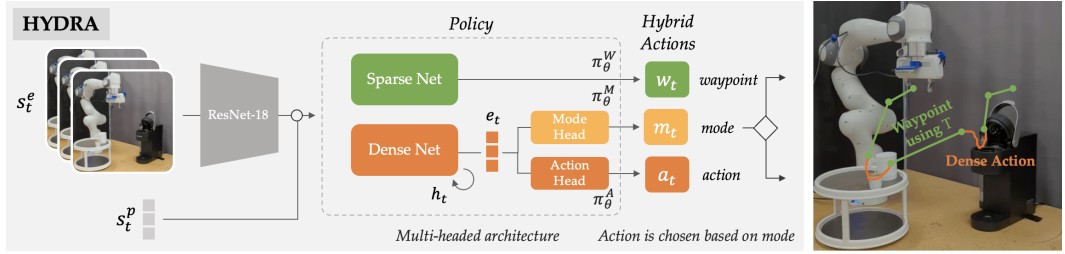

Figure 1: Multi-headed architecture of HYDRA: During training, we learn to predict waypoints, low level actions, and the mode label for each time step. One network (Dense Net) predicts the low level action $a_t$ and the mode $m_t$; both the action and mode heads of Dense Net share an intermediate representation $e_t$. A separate network (Sparse Net) predicts the high level waypoint $w_t$. At test time, we sample $m_t$ and either servo to reach a waypoint ($m_t = 0$) without requerying the policy, or follow a dense action for one time step ($m_t = 1$). An example of how sparse and dense modes can be arbitrarily stitched together at test time is shown on the right.

## 4    HYDRA: A Hybrid Action Representation

To reduce distribution shift, our insight is that most robot manipulation tasks are a combination of *sparse* waypoint-reaching, such as reaching for an object or lifting a mug towards a shelf, and *dense* low-level actions, such as grasping an object or balancing a mug stably on a shelf. Waypoints capture free-space motions but struggle to capture dexterous or precise behaviors. Conversely, low-level actions capture these dynamic behaviors but are often redundant during long free-space motions.

Instead of learning from only velocities or waypoints, HYDRA learns a *hybrid action representation* consisting of both high-level waypoints in the robot's proprioceptive space $w \in \mathcal{P}$ and low-level actions $a \in \mathcal{A}$. Additionally, we learn to dynamically switch between these modes by predicting which mode $m \in \{0, 1\}$, sparse or dense, should be executed at each demonstrated state. Mode labels are annotated with little extra cost by experts either during or after data collection. This flexible abstraction leads to (1) a compressed action space that reduces compounding errors without sacrificing dexterity or generality, and (2) a more consistent, simple low-level action distribution through action relabeling during the sparse periods. This section presents an overview of the approach, followed by discussions on mode labeling, action relabeling, and training/testing procedures.

**Overview**: The multi-headed architecture of HYDRA is outlined in Fig. 1, with heads $\pi_\theta^M : \mathcal{O} \to \{0, 1\}$, $\pi_\theta^A : \mathcal{O} \to \mathcal{A}$, $\pi_\theta^W : \mathcal{O} \to \mathcal{P}$, for mode, action, and waypoint respectively. One network, Dense Net, predicts the low-level action $a_t$ and the mode $m_t$ at each input $o_t = \{s_t^e, s_t^p\}$. Another network, Sparse Net, separately outputs the desired *future* waypoint $w_t$ for input $o_t$. We assume waypoints can be reached using a known controller $\mathrm{T} : \mathcal{O} \times \mathcal{P} \to \mathcal{A}$ which converts the state and desired waypoint into a low-level action (e.g. a linear controller, see the right side of Fig. 1). In practice, Dense Net is recurrent since both the mode and action are highly history-dependent. Sparse Net in contrast only uses the current observation, since waypoints are less multi-modal and history dependent than actions. Then at test time, HYDRA predicts the mode $m_t$ and follows the controller T until reaching the waypoint during predicted sparse periods, and follows low-level actions at each step during predicted dense periods. See Appendix C for more details.

### 4.1    Data Processing: Mode Labeling and Action Consistency

To dynamically switch action abstractions, we need labeled modes $m_t$, waypoints $w_t$, and actions $a_t$ at each time step. We first obtain binary mode labels $m_t$ from humans, and then use the mode labels to extract waypoints and to relabel low-level actions. Importantly, modes can be labeled either during demonstration collection (e.g. with a simple button interface), or entirely after demonstration collection (e.g., labeling each frame with its mode). With modes labeled, we can segment each demonstration into sparse waypoint and dense action phases. We provide the details of the labeling and segmentation process in Appendix B. For each sparse phase, we can extract the desired future waypoint $w_t$ at $o_t$: if $m_t = 0$ (sparse), the future waypoint is final proprioceptive state $w_t = p_{t'}$ in that sparse segment, where $t' > t$. But if $m_t = 1$ (dense), the waypoint is the next proprioceptive state $w_t = p_{t+1}$. This yields a dataset of $\hat{\mathcal{D}}$ of $(o, a, w, m)$ tuples. Now the policy has full supervision to learn the modes, waypoints, and actions.

**Mode Labeling Strategy**: Since waypoints will be reached online with controller T, the main requirement for labeling modes is that during sparse phases ($m_t = 0$), the labeled waypoint $w_t$ should be reachable via T starting from $o_t$ (i.e., without collision): for example, if the demonstrator starts in free space and labels a waypoint close to coffee K-pod, and if the policy uses a linear P-controller as T, then the K-pod waypoint should be reachable from the initial pose in a straight-line path. Otherwise, the learned policy might collide when it tries to reach similar waypoints. We do not assume access to a collision-avoidance planner as T in this work, but if one has access to a planner then T can always reach the desired waypoint, so this reachability requirement can be ignored. Other considerations for mode labeling and a discussion of mode sensitivity is provided in Appendix B. We specifically show that our method is not overly sensitive to mode labeling strategies outside of the collision-free requirement above. Furthermore, we show that mode labels can be learned from substantially fewer examples without a major effect on performance Appendix D.3.

**Relabeling Low-Level Actions**: As discussed in Section 3, action consistency can improve policy performance by simplifying the BC objective in Eq. (1) and thus reducing $||\epsilon_t||$, provided the data has enough state coverage. However, making actions consistent during data collection is challenging and can often reduce state coverage [22], so instead HYDRA performs *offline* action relabeling, i.e., after collection. To relabel human actions $a_t$ during the sparse periods, HYDRA uses waypoint controller T to recompute a new action at each demonstrated robot state $s_t^p$ based on the waypoint $w_t$. We lack a consistent relabeling strategy for dense periods, so we leave this to future work.

However, a subtle challenge with offline relabeling is that changing the actions in the data can put the policy out of distribution at test time, since new actions can lead to new states online. For example, if an arc path was demonstrated to get to a waypoint, but a linear controller is used for relabeling, the linear action will take us off that path. HYDRA avoids this problem by using a waypoint controller T online during sparse periods, meaning relabeled actions will not be deployed online. Rather, this action relabeling serves primarily to simplify the dense action learning objective of HYDRA and increase action consistency in the overall dataset.

A natural question arises: since sparse actions will be executed with T online, could we instead further simplify learning by avoiding training on dense actions during sparse periods? If HYDRA mispredicts a sparse mode as dense, then the dense actions will still be executed online, so HYDRA should still be trained on dense actions during sparse periods as a back-up. We show that reducing the training weight of dense actions during sparse periods hurts performance in Appendix D.5.

### 4.2 Training and Evaluation

**Training:** HYDRA is trained to both imitate low-level actions $a$ with policy $\pi_\theta^A$, high-level waypoints $w$ with $\pi_\theta^W$, and the mode $m$ with $\pi_\theta^M$ at each time step. To balance the waypoint and action losses, we use a mode-specific loss at each time step that weighs the current mode's loss with $(1-\gamma)$, and the other mode's loss with $\gamma$. Given a processed dataset $\hat{\mathcal{D}}$ consisting of tuples of $(o, a, w, m)$, we modify the loss in Eq. (1) with the new heads of HYDRA (mode, action, and waypoint):

$$\mathcal{L}_a(\theta) = -\mathbb{E}_{(o,a,w,m) \sim p_{\hat{\mathcal{D}}}} \left[ (1 - \alpha_m) \log \pi_\theta^A(a|o) + \alpha_m \log \pi_\theta^W(w|o) \right] \tag{2}$$

$$\mathcal{L}_m(\theta) = -\mathbb{E}_{(o,a,w,m) \sim p_{\hat{\mathcal{D}}}} \left[ m \log \pi_\theta^M(m = 1|o) + (1 - m) \log \pi_\theta^M(m = 0|o) \right] \tag{3}$$

$\mathcal{L}_a$ weighs the BC loss for waypoints and actions by the current mode: $\alpha_m = m\gamma + (1-m)(1-\gamma)$ is the mode-specific weight for the sparse waypoint part of $\mathcal{L}_a$. If we are in sparse mode ($m = 0$), then $\alpha_m = 1 - \gamma$, but in dense mode, $\alpha_m = \gamma$. Thus, a low gamma encourages the model to fit the loss for the current mode *more* than the loss for the other mode, and $\gamma = 0.5$ will be a mode-agnostic weighting. See Appendix D.5 for results of ablating $\gamma$. $\mathcal{L}_m$ is the mode cross entropy classification loss. Combining these terms with mode loss weight $\beta$, we get the full HYDRA objective:

$$\mathcal{L}(\theta) = \mathcal{L}_a(\theta) + \beta \mathcal{L}_m(\theta) \tag{4}$$

**Evaluation:** During evaluation, the policy chooses the mode using $\tilde{m}_t$. If $\tilde{m}_t = 0$, the model will servo in a closed-loop fashion to the predicted waypoint $\tilde{w}_t$ using controller T. The policy is queried at every step to continually update the policy hidden state, but importantly its outputs are ignored until we reach the waypoint to avoid action prediction errors. If $\tilde{m}_t = 1$, the model will execute just the next step using the predicted dense action $\tilde{a}_t$.

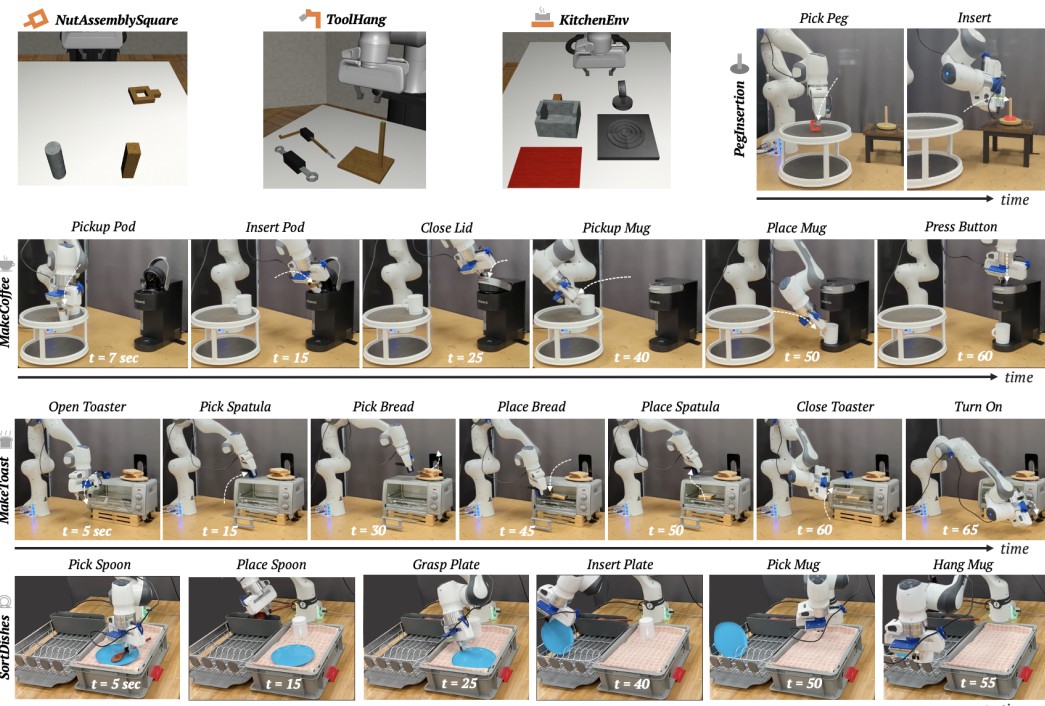

Figure 2: Simulation & Real-world environments, with task stages shown for real world tasks. **Simulation**: In *NutAssemblySquare*, we pick up a square nut at various positions and orientations and insert it onto a vertical square peg. In *ToolHang*, a hanging frame is inserted onto a fixed stand, followed by placing a tool on the frame. Both the frame and tool poses are randomized. Frame insertion is challenging due to the small insertion area. *KitchenEnv* involves turning on a stove, moving a pot onto the stove, putting an object in the pot, then moving the pot to a serving area. **Real World**: *PegInsertion* involves inserting a peg with a hole in the center onto a round insertion rod (top right); the peg location and geometry are varied. *MakeCoffee* is a 6-step task (top middle row) involving picking up a K-pod, inserting it into a Keurig machine, closing the lid of the Keurig, positioning a mug, and then pressing start on the Keurig; the K-pod location and mug orientations are varied. Unlike prior work [13], we include a mug component. *MakeToast* has 7-steps (bottom middle row): a hinged toaster oven is opened, a spatula is picked up, bread is placed in the toaster, the toaster is closed, and the dial is turned to start. Bread and spatula initial poses vary. *SortDishes* (bottom row) has 6 stages: pick up spoon, place spoon in rack, grasp plate and insert it into rack, and grasp mug and hang the mug. All objects poses vary.

## 5 Experiments

We evaluate the performance of HYDRA in 3 challenging simulation environments and 4 complex real world tasks, shown in Fig. 2. These tasks cover a wide range of affordances and levels of precision, from precisely inserting a coffee pod to picking up bread with a spatula. See Appendix C for model hyperparameters, data collection, and training details. Videos can be found on our website.

**Data Collection**: We leverage proficient human demonstration data for simulated tasks from robomimic [19]. Mode labels and waypoints were annotated offline for simulation datasets as described in Appendix B. Demonstrations for real world tasks were collected by a proficient user using VR teleoperation using an Oculus Quest 2. Mode labels and waypoints were provided during data collection using the side button on the Quest VR controller with no added collection time. All methods share the same underlying dataset.

**Simulation**: In Fig. 3 (top row), we compare our method to BC and BC-RNN for the *NutAssemblySquare* and *ToolHang* tasks (state-based), as well as the *KitchenEnv* task (vision-based) from robosuite (see top row in Fig. 2). Our method improves performance on the *NutAssemblySquare* task, where baselines are already quite strong. We also ablate the data size from 200 demos to 100 and 50 in Fig. 3, illustrating that HYDRA is more sample efficient than baselines, with the gap growing as data size decreases. HYDRA-NR in Fig. 3 removes action relabeling and drops performance by 8%, which we attribute to high action multi-modality in non-relabeled sparse periods, but for 50 demos it performs similarly, likely due to high action consistency just from having less data.

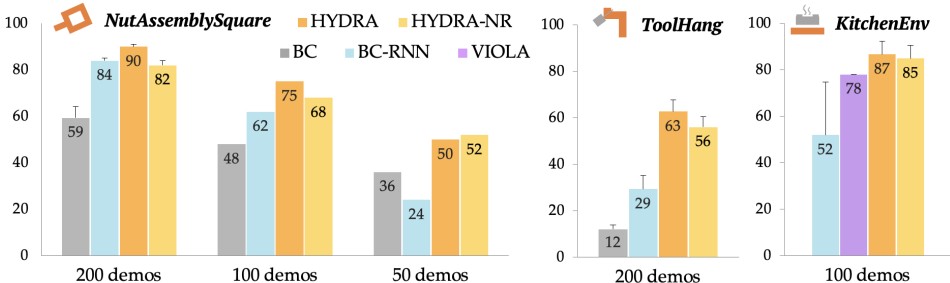

Figure 3: Sim Results for HYDRA vs. BC, BC-RNN, and VIOLA: best checkpoint success rate averaged over three seeds. **Left to Right**: *NutAssemblySquare* (state), *ToolHang* (state), and *KitchenEnv* (vision) tasks. HYDRA beats baselines on all of these tasks, and even beats VIOLA [13] on the kitchen task despite using a much smaller and simpler model. We also show a comparison for BC-RNN and HYDRA with decreasing data sizes for *NutAssemblySquare*, showing that our method is more sample efficient than BC-RNN. HYDRA without action relabeling (HYDRA-NR, *NutAssemblySquare* and *ToolHang*) drops performance by 7-8%.

For *Tool Hang* (top middle in Fig. 3), which is long horizon, consisting of many waypoint / dense periods and requiring much higher precision, the gap is even bigger from HYDRA to BC and BC-RNN. While the best baseline gets 29%, our method reaches 63% with the same inputs. Again, no action relabeling (HYDRA-NR) drops performance by 7% but is still much better than baseline.

For *KitchenEnv* (vision-based), we also compare to VIOLA [13], an image-based model that uses bounding box features and a large transformer architecture to predict actions. Once again, HYDRA is able to outperform BC-RNN by 35% on this long horizon task. HYDRA also outperforms VIOLA by 9%, despite using a simpler and smaller model.

In Appendix D, we discuss HYDRA-NR results, show a waypoint-only baseline, mode labeling ablations, and a relabeling-only ablation where action consistency is improved but the waypoint controller is not used online. In Appendix D.3, we show that mode labels can be learned with fewer examples without large performance drops (using 25% of mode labels drops performance by 10%).

**Real World**: In Fig. 4, we compare our method to BC-RNN (vision-based) for four high precision tasks: *PegInsertion*, *MakeCoffee*, *MakeToast*, and *SortDishes*. The latter three are long-horizon, and Fig. 4 shows cumulative success per task stage. In *PegInsertion*, our method substantially outperforms BC-RNN at both peg grasping and precise insertion portions of the task, thanks to combining precise waypoints with flexible low level actions where necessary.

For *MakeCoffee*, HYDRA once again beats BC-RNN and VIOLA by a substantial margin at all task stages. All methods perform well in grasping the K-pod, but the performance of the baselines declines rapidly in the following phases. While BC-RNN failed to do this task in prior work, we see with a bit of parameter tuning, BC-RNN is a strong baseline, achieving 20% performance [13]. VIOLA's reported performance in prior work for pod insertion and closing the lid is 60%, matching what we observe for the corresponding stage of our coffee task. Our task adds two more stages (picking up and placing a mug before pressing the button), interestingly causing the final success rate of VIOLA to drop to 20%, the same as BC-RNN. Using the same parameters and model size as BC-RNN, HYDRA achieves 80% final success at this task with the same underlying dataset.

For *MakeToast* and *SortDishes*, HYDRA performs better on all stages of the task as compared to BC-RNN. We omit VIOLA in these tasks since, as seen in the coffee task, BC-RNN is a competitive baseline (see Appendix A). Both tasks consists of several bottleneck stages where performance drops sharply. In *MakeToast*, for picking up bread, the spatula must slide underneath a bread slice – HYDRA passes this stage 70% of the time, beating BC-RNN by 30%. The last stage (turning the toaster on) is particularly challenging for all methods (precise dial grasp, partial observability), but HYDRA completes it 20% of the time compared to 0% for BC-RNN. In *SortDishes*, the final hang-mug stage similarly requires high precision (small space between rack and mug hang point, partial observability). Not including the challenging last stage, HYDRA beats BC-RNN by 40% on this task. See Appendix D.1 for rollouts of each task for each model.

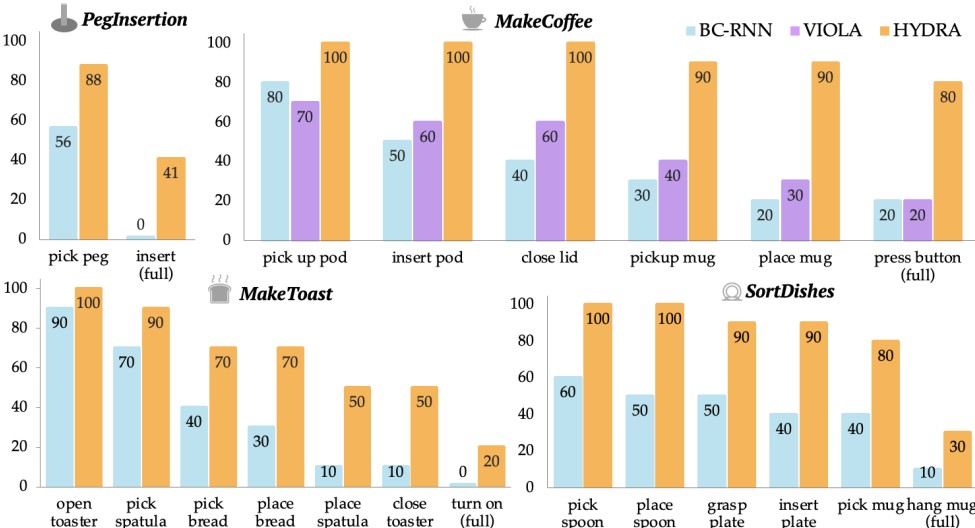

Figure 4: Real Results for HYDRA vs. BC, BC-RNN, and VIOLA. The x-axis denotes each stage (right-most value is the final success rate). **Top Left**: HYDRA vs. BC-RNN on the real *PegInsertion* task for 50 demos under 32 rollouts across 4 different nuts. This task requires very precise grasping and insertion of multiple types of nuts, which our method does with high success. While baseline is unable to perform insertion, HYDRA gets 41% success. **Top Right**: *MakeCoffee* long-horizon task for 100 demos under 10 rollouts. Our method beats baseline by 60%. **Bottom Left**: *MakeToast* long-horizon task for 100 demos under 10 rollouts. While both methods struggle to turn the toaster on, HYDRA is able to reach 50% success for 6/7 stages compared to 10% for baseline. **Bottom Right**: *SortDishes* for 100 demos under 10 rollouts. Waypoints in HYDRA precisely capture the diverse poses in this task, beating BC-RNN by 40% and 20% for the last two stages.

We observe that the performance gain for HYDRA in our real world experiments is notably higher than in simulation. We hypothesize this is due to (1) higher variance in action playback on the real robot setup, which HYDRA mitigates during sparse periods using the closed-loop waypoint controller, and (2) increased potential for compounding errors in longer tasks. Overall, HYDRA is well-suited to long horizon tasks even with many high-precision bottleneck stages, due to its ability to switch between waypoints and dense actions and its ability to increase action consistency offline. We also observed that in our real world tasks, HYDRA exhibits emergent retrying behavior, often re-servoing to a consistent and in-distribution waypoint to retry a failed dense period.

## 6 Discussion

**Summary**: In this work, we propose HYDRA, which uses a flexible action abstraction to reduce compounding errors, and improves action consistency while maintaining the state diversity present in uncurated human demonstrations. HYDRA learns to dynamically switch between following waypoints and taking low level actions with a small amount of added mode label supervision that can be provided either online or offline. HYDRA substantially outperforms baselines on three simulation tasks and four real world tasks that involve long horizon manipulation with many bottleneck states.

**Limitations & Future Work**: While only a minor amount of added supervision, HYDRA relies on having expert-collected mode labels. We show that mode labels can be learned from much less data in Appendix D.3, but future work might consider using unsupervised methods for mode labeling, e.g., skill segmentation [39] or automatically extracting "linear" portions of a demonstration. We also hypothesize multi-task datasets can help learn a general mode-predictor that can be fine-tuned or deployed zero-shot on novel tasks. Furthermore, when mode labels are collected online, mode labeling can add a mental load for the demonstrator and might also influence the quality of the data on its own. Future work might conduct more extensive user studies to better understand the effect of providing mode labels for both the demonstrator and the final learning performance.

Despite these limitations, HYDRA is a simple and easy-to-implement method, and it is exciting that it shows substantial improvement over state-of-the-art imitation learning techniques and significant promise in solving challenging manipulation tasks in the real world.

## Acknowledgments

This research in advancing imitation learning was supported by ONR, DARPA YFA, Ford, and NSF Awards #1941722, #2218760, and #2125511.

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

We provide a broader discussion of our method in this appendix. In Appendix A we list a set of *motivating* questions that may arise during reading the main text of this work and provide our response with links to additional details in corresponding sections in the Appendix. In Appendix B we discuss how to collect mode labels, and considerations for how to define waypoint and dense segments. In Appendix C, we outline training procedures, model architectures, and hyperparameters. In Appendix D, we provide ablation experiments for our method, including sensitivity to mode labels, learning mode labels from less data, ablations to $\gamma$, and robustness of HYDRA to added system noise.

## A   Motivating Questions

**Intuitively, why does HYDRA help improve BC?** Humans demonstrate manipulation tasks at an abstraction level that is different from how the robot interprets the data. A BC agent interprets the data *literally* as taking a specific action at an exact state while the human is *noisely* reaching for an object. At the high level, HYDRA improves BC by realigning the task abstraction of the robot to the human demonstrator during waypoint mode of the task. Concretely, HYDRA curates the dataset in a way that improves action consistency and optimality without reducing state diversity and hence allowing the learned policy to stay closer in distribution at test time.

**What's the relationship of HYDRA with works in hindsight relabeling?** Hindsight relabeling [40] is the idea of relabeling past experiences of goal-reaching trajectories with the final state it reaches to reuse any sub-optimal data (especially for reinforcement learning settings). Recent work of Zhang and Stadie [41] draws the connection between goal-conditioned imitation learning and hindsight relabeling from a divergence-minimization perspective. The current implementation of HYDRA operates in single-task imitation learning setting, and therefore is only remotely related to the idea in hindsight relabeling. From this perspective, one can think of HYDRA as effectively reducing divergence of the dataset's action distribution by relabeling actions for the waypoint periods of the trajectory.

**Does mode labeling during collection change demonstrator behavior?** We explain the mode labeling process during collection in Appendix B. We acknowledge that asking the demonstrator to provide mode labels during data collection adds additional cognitive load during demonstrating the task, and at the same time may change their demonstration behavior. In practice, asking the demonstrator to provide the two mode labels can communicate the structure the robot leverages to learn tasks and may in turn allow the human to provide better demonstrations (such as consistent waypoints etc.). However, we leave this user study to future work.

**How sensitive is HYDRA to mode labeling?** In our experiments, we (experts in this task) provided the mode labels for different tasks. We found HYDRA to be robust to the labeling strategies across the two labelers. For simulated environments, we use existing datasets and labeled the modes using an interface that shows the robot view of the task and the human annotator marks whether a frame is waypoint or dense mode. For real robot tasks, the human demonstrator provides the mode labels as they provide the demonstration using a button on the teleoperation controller. We provide guidelines for how to perform mode labeling in Appendix B.

**How were baselines chosen?** BC was not included in Kitchen or the real world tasks since it was substantially lower performing than BC-RNN, both in our state-based simulated results and in some initial testing on visual domains. For VIOLA, as noted in the text, BC-RNN was shown to be a strong baseline in the MakeCoffee task, nearly matching VIOLA's performance despite using a much smaller network architecture. Secondly, VIOLA takes longer to both train and perform inference due to the use of a region proposal network and a large transformer architecture. Thirdly, VIOLA changes the *input* and encoding structure on top of an approach like BC-RNN, whereas HYDRA changes the *output*. Thus these methods are actually compatible with each other, and the point of including VIOLA was not as a primary baseline but to show that designing intelligent *output action* spaces can be as or more beneficial than large changes in the *input features* or encoding structure. This is on top of the fact that HYDRA also uses substantially fewer parameters than VIOLA, and thus BC-RNN is a more directly comparable approach. Given these limitations,

and our strong BC-RNN baselines, we argue the addition of VIOLA does not provide any essential signal about HYDRA.

## B    Labeling Modes in HYDRA

### B.1    Providing Mode Labels

The primary assumption made in HYDRA is the availability of mode labels for sparse and dense periods. Here we provide a discussion of how mode labels can be collected via a simple binary "click" interface, either during demonstration collection or after collection. In either case, we can label dense periods and exact waypoints using a single binary "click" variable via an external button: to label a waypoint at the end of a sparse period, we provide a single click at the waypoint state; to label a dense period, we sustain the click until the end of the dense period (see left image in Fig. 5). Once clicks are labeled, we demarcate periods in between clicks as sparse modes, and periods with sustained clicks as dense modes (see right image in Fig. 5).

**Uncurated demonstration**             **Relabeled demonstration**

Figure 5: Mode labeling example for peg-insertion task. For each demo a human labels binary click signals at each time step (labeled during or after collection) to segment trajectories into arbitrary sequences of sparse waypoint phases and dense action phases. **Left**: Uncurated demo, with single clicks and sustained clicks shown. **Right**: Relabeled demo, with waypoint and dense segments overlayed in green and orange, respectively. We also relabel actions for the states in sparse segments with the optimal waypoint reaching action shown in white. For sparse segments, the waypoint head of HYDRA is trained to output the final waypoint at each state along the trajectory.

With the trajectories segmented into sparse and dense modes, we can extract the desired future waypoint $w_t$ for each $o_t$: if $m_t = 0$ (sparse), the future waypoint is the next labeled "single click" proprioceptive state $w_t = p_{t'}$ where $t' > t$ (for example, states $o_t$ with $t_1 \leq t < t_2$ in Fig. 5 will use $w_t = p_{t_2}$). But if $m_t = 1$ (dense), the waypoint is the next proprioceptive state $w_t = p_{t+1}$. Thus we construct a dataset of $\hat{\mathcal{D}}$ of $(o_t, a_t, w_t, m_t)$ tuples. Now the policy has full supervision to learn both the action and waypoint as well as the mode of operation. In Algorithm 1, we outline this process of turning a click-labeled dataset into per-step waypoints and mode labels.

### B.2    Waypoint Controller

For all experiments in the main text, we use a linear controller $\mathrm{T_{linear}}$ for reaching waypoints online. This means that when HYDRA predicts a waypoint period ($\tilde{m}_t = 0$), it will servo closed loop until

---

**Algorithm 1** Labeling Modes

---
1: Given click-labeled dataset: $D = \{(o_t, a_t, c_t) \dots\}$
2: $\hat{D} = \{\}$
3: **for all** t **do**
4:     $m_t = c_t \ \& \ (c_{t-1} \mid c_{t+1})$                            ▷ Sustained click for dense
5:     *// Mark single click as waypoint*
6:     isolated $= \neg c_{t-1} \ \& \ c_t \ \& \ \neg c_{t+1}$
7:     *// Mark start of dense period as waypoint*
8:     start_dense $= \neg c_{t-1} \ \& \ m_t$
9:     **if** isolated **or** start_dense **then**
10:         $w_{t_p:t-1} = p_t$                               ▷ Set previous waypoints
11:         $t_p = t$                                     ▷ start of next sparse phase
12:     **else if** $m_t = 1$ **then**
13:         *// During dense mode the next state is a waypoint*
14:         $w_t = p_{t+1}$
15:         $t_p = t$
16:     Add $(o_t, a_t, w_t, m_t)$ to $\hat{D}$

---

it reaches the predicted $\tilde{w}_t$ or times out after $N$ seconds. In all of our experiments, the waypoint follower times out after $N = 5$ seconds if it has not reached the waypoint.

For this closed loop servoing during test time, the policy will still be *called*, but its outputs will be ignored. This is important for recurrent models specifically (e.g., Dense Net), since the hidden state for the policy should be updated similarly to how it was trained (on all states, even during the sparse period). While this mitigates the changes in the hidden state, this might still induce a different hidden state than was produced offline, since the human policy followed a non-optimal path to reach waypoint $w$ from state $s_t$, as compared to the optimal online trajectory generated by T. For example, if the demonstrator follows an arc-like trajectory to pick up a coffee pod and marks the waypoint right before picking up the coffee pod, then online the policy with $T_{\text{linear}}$ will servo to that waypoint directly; the hidden state for these two paths will likely be different. This problem is difficult to observe in practice, and did not empirically show up in practice (as evidenced by the improved performance of our method compared to baselines).

In theory, one could bypass this issue by "skipping" the hidden state of the policy over entire sparse segments during training. Then during test time, if the policy outputs $\tilde{m}_t = 0$, the policy would not be called again until reaching the output $w_t$. However, this requires loading entire sparse segments and more in the training batches, which is computationally expensive and less simple then loading batches of fixed horizon as is commonly done. We leave a broader analysis of the hidden state problem for future work.

Additionally, we experimented with several controller gains and did not notice any effect on performance. Therefore we choose a fast controller to reach waypoints. These gains are constant for all experiments.

For all experiments, we use Polymetis for control [42], with 10Hz policy frequency and 1kHz controller frequency. We use a Hybrid Joint Impedance controller to follow end effector waypoint and velocity commands, but our method is not tied to any one underlying controller. End effector pose interpolation follows the shortest position and orientation path in SO(3).

### B.3 Mode Labeling Sensitivity

In our experiments, we noticed that mode labeling was quite robust to different labeling strategies provided that the labeling strategy satisfies the following guidelines.

**Waypoint Following Behaviors**: Waypoint following behaviors should be labeled for free-space motions in the environment, when the robot is "in transit" (e.g., reaching). As described in Section 4.1, a key consideration for mode labeling is making sure labels for sparse periods are compatible with the waypoint controller T. For example, if we are following a linear controller, waypoint

segments should be reproducible with straight line segments from any start state along the waypoint segment. For a given $(m_t, w_t, s_t)$, then if $m_t = 0$, we should be able to reach waypoint $w_t$ from $s_t$ with T (i.e. without timing out). As mentioned in the main text, if T includes collision avoidance as part of the controller, then we no longer have any requirements on waypoint following behaviors.

An important yet subtle point here is that even though a linear controller requires that the path between any two waypoints is collision free, we can always arbitrarily add more waypoints to avoid collision. Since SparseNet in our framework is visually aware policy that is learned on each transition in the data (i.e. a closed loop policy), it can learn to sequence together multiple waypoints in a row without any additional cost. We show that multiple waypoints can be sequenced in a row in Appendix D.2, and several of our experiments involve "pre-grasp" waypoints to avoid collision in the scene (e.g. *NutAssemblySquare* to avoid collision between the nut and the peg).

**Dense Object Interaction**: Dense periods should include (but is not limited to) all object interactions in the scene where "collision" with the scene is necessary (e.g., grasping a coffee pod, inserting the coffee pod into the coffee machine, picking up toast with a spatula). Humans excel at identifying these types of interactions, so these segments are quite easy to label. The exact amount of time "padded" onto these dense periods did not seem to affect learning in our experiments. Note that if each entire demo is treated as a dense period, our algorithm reduces to BC.

**Labeling Strategy Consistency**: The final consideration is for the consistency of the mode labeling strategy *between* different demonstrations. Variation in the exact boundaries / choices for waypoints and dense segments is inevitable with human labeling. While the effects of certain types of variation can be quite difficult to quantify in general, we believe that is important to minimize this variation without adding additional burden on the user. In our experiments, for each task and dataset, we have only one user provide the mode labels, according to a single strategy. For example, in the *NutAssemblySquare* task, where to goal is to insert a square nut onto a peg, a user might define the following strategy:

1. Reach waypoint above the square nut (sparse)
2. Go down, grasp, pick up (dense)
3. Move the nut up (sparse)
4. Move the nut above the insertion point (sparse)
5. precisely insert the nut on the peg (dense)

In general our method is quite robust to variations within a single mode labeling strategy (for a single labeler), and we do no additional post-processing on mode labels or waypoints in any of our experiments.

### B.4 Training on Mode Labels

With labeled modes and waypoints, HYDRA learns to predict the mode, the waypoint, and the low-level action at every time step according to the loss in Eqn. 4. However, due to training a higher dimensional action space (e.g. for robot poses: $|\mathcal{A}| = 7 + 7 + 1$) with a supervised objective, over-fitting can be a key concern during training. For all vision-based experiments, we perform random cropping to 90% the image size. However, there are several interesting mode-specific augmentations that can be done using mode labels and waypoints to mitigate this problem:

**Mode Smoothing**: While the simple binary cross entropy mode loss in Eqn. 3 suffices for learning to predict modes, sometimes the hard boundary between segments can lead to mode oscillation or cycling when evaluating at test time. For example, model might predict a dense mode, then predict a sparse mode at the next step that brings it back to the previous state, and repeat. In these cases (which are rare in practice) it can be beneficial to *smooth* the mode labels to extract continuous probabilities for the mode label at each step: $p(\tau_m) = \text{convolve}(\tau_m, [\frac{1}{n}, ... \frac{1}{n}])$, where $n$ is the kernel size. This yields the following loss:

$$\mathcal{L}_m(\theta) = -\mathbb{E}_{(o,a,w,m)\in\hat{\mathcal{D}}} \left[ p(m) \log \pi_\theta^M (m = 1|o) + (1 - p(m)) \log \pi_\theta^M (m = 0|o) \right] \quad (5)$$

With this smoothing of the mode labels, we are effectively removing the hard boundary between sparse and dense periods, which can help generalization for the mode prediction head of HYDRA at test time.

**Waypoint Period Augmentation**: It is common in the literature to add small amounts of proprioceptive state noise (increasing state diversity) to demonstrations. However, during object interaction (i.e. dense periods), this noise can make policy learning more difficult since minor variations in the state can have large changes in the action space. However, with knowledge of sparse and dense modes in HYDRA, we could add diverse state augmentations to the proprioceptive state during only the sparse periods. This waypoint period augmentation can help reduce overfitting in SparseNet, since we will learn to reach the same waypoint (action) from many different robot poses (state).

Both mode smoothing and waypoint augmentation, while not utilized in our experiments, illustrate the potential for new augmentation strategies that arise with access to mode labels.

| Environment | Method | # Demos | $B$ | $H$ | lr | $\gamma$ | $\beta_m$ | $\lvert i \rvert$ | $\lvert D \rvert$ | $\lvert S \rvert$ | $\lvert \pi_\theta^A \rvert$ | $\lvert \pi_\theta^M \rvert$ | GMM |
|---|---|---|---|---|---|---|---|---|---|---|---|---|---|
| *NutAssemblySquare* | BC | 200 | 256 | – | 1e-4 | – | – | – | 400 | – | – | – | 0 |
| | BC-RNN | 200 | 256 | 10 | 1e-4 | – | – | – | 400 | – | – | – | 0 |
| | HYDRA | 200 | 256 | 10 | 1e-4 | 0.5 | 0.01 | – | 400 | 200 | 200 | 200 | 0 |
| *ToolHang* | BC | 200 | 256 | – | 1e-4 | – | – | – | 400 | – | – | – | 5 |
| | BC-RNN | 200 | 256 | 10 | 1e-4 | – | – | – | 1000 | – | – | – | 5 |
| | HYDRA | 200 | 256 | 20 | 1e-4 | 0.5 | 0.1 | – | 1000 | 400 | 400 | 400 | 0 |
| *KitchenEnv* | BC-RNN | 100 | 16 | 10 | 1e-4 | – | – | 64 | 1000 | – | – | – | 5 |
| | HYDRA | 100 | 16 | 10 | 1e-4 | 0.5 | 0.01 | 64 | 1000 | 400 | 400 | 400 | 5 |
| *PegInsertion* | BC-RNN | 75 | 8 | 10 | 1e-4 | – | – | 64 | 1000 | – | – | – | 0 |
| | HYDRA | 75 | 8 | 10 | 1e-4 | 0.5 | 0.01 | 64 | 1000 | 1000 | 1000 | 1000 | 0 |
| *MakeCoffee* | BC-RNN | 100 | 8 | 10 | 1e-4 | – | – | 64 | 1000 | – | – | – | 0 |
| | HYDRA | 100 | 8 | 10 | 1e-4 | 0.5 | 0.01 | 64 | 1000 | 1000 | 1000 | 1000 | 0 |
| *MakeToast* | BC-RNN | 80 | 8 | 10 | 1e-4 | – | – | 64 | 1000 | – | – | – | 0 |
| | HYDRA | 80 | 8 | 10 | 1e-4 | 0.5 | 0.01 | 64 | 1000 | 1000 | 1000 | 1000 | 0 |

Table 1: Hyperparameters for each environment, from left to right: $B$ is batch size, $H$ is the horizon length for training, lr is the learning rate, $\gamma$ is the per time step weighting of the current mode, $\beta_m$ is the weighting of the mode loss, $\lvert i \rvert$ is the image encoding size (for each image), $\lvert D \rvert$ is the hidden-size for recurrent dense networks (DenseNet, BC-RNN) or the MLP width (BC), $\lvert S \rvert$ is the width of the SparseNet MLP (3 layers), $\lvert \pi_\theta^A \rvert$ is the width of the action head (2 layers), $\lvert \pi_\theta^M \rvert$ is the width of the mode head (2 layers), and finally GMM is the number of Gaussian mixtures (or 0 if deterministic) used for the dense action space. The top 3 rows are sim environments, where the first two are state only. The bottom three rows are vision-based real-world experiments. Hyperparameters stay mostly constant for HYDRA between experiments, with larger policy sizes for harder tasks. In almost all cases, BC-RNN, BC, and HYDRA share the same hyperparameters.

## C  Model Architectures & Training

To train HYDRA, we use a similar procedure as in prior work [19, 14]. For each input of shape $D_1 \times \dots D_N$, we load sequential batches of size $B \times H \times D_1 \times \dots D_N$, where $H$ is the horizon length. Next we outline the network design for HYDRA, and hyperparameters used in each environment.

### C.1  Network Design

As described in Section 4, HYDRA consists of SparseNet, which predicts the waypoint trajectory $\tau_w$, and DenseNet, which predicts the mode trajectory $\tau_m$ and low level action trajectory $\tau_a$. Both networks condition on the same input observation space (proprioceptive state trajectory $\tau_{s^p}$ and environment state $\tau_{s^e}$). For vision based experiments, $s^p$ consists of both wrist mounted and external camera observations. Each image is encoded via a ResNet18 architecture encoder (two encoders, $E_\theta^{\text{ext}}$, $E_\theta^{\text{wrist}}$, with separate parameters) which is trained end-to-end. Next, the image encodings are concatenated along with the proprioceptive trajectory $\tau_{s^p}$.

### C.2  Model & Training Details

Visual encoders use a ResNet-18 architecture trained end-to-end on both external images and end-effector images. We train all methods for 500k training steps over 3 random seeds, and like prior

**Algorithm 2** Training HYDRA

1:  Given $N$ (number of training steps)
2:  Given mode-labeled dataset: $\hat{D} = \{(\tau_o, \tau_a, \tau_w, \tau_m) \dots\}$
3:  Networks $E_\theta^{\text{ext}}, E_\theta^{\text{wrist}}, \pi_\theta^W, \pi_\theta^A, \pi_\theta^M$
4:  **for** $i$ **in** range($N$) **do**
5:      $\tau_o, \tau_a, \tau_w, \tau_m \sim \hat{D}$            $\triangleright$ Load $(B \times H \times \dots)$
6:      $\tau_i = E_\theta^{\text{wrist}}(\tau_o) \oplus E_\theta^{\text{ext}}(\tau_o) \oplus \tau_{s^p}$        $\triangleright$ Encode
7:      $\tau_w = \pi_\theta^W(\tau_i)$          $\triangleright$ waypoint (SparseNet)
8:      $\tau_m = \pi_\theta^M(\tau_e)$          $\triangleright$ mode (DenseNet)
9:      $\tau_a = \pi_\theta^A(\tau_e)$          $\triangleright$ action (DenseNet)
10:     Compute $\mathcal{L}(\theta)$ in Eqn. 4 and update $\theta$

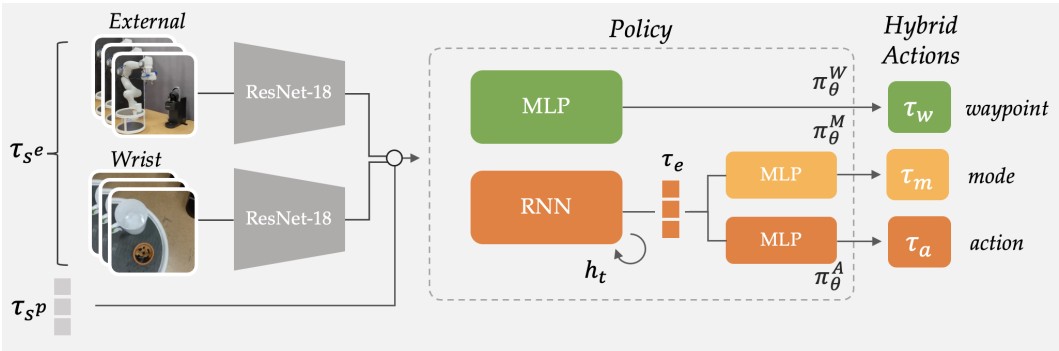

Figure 6: Specific instantiation of HYDRA for vision based experiments.

work we report the average over the best performing checkpoints per run [19]. We found that BC policy performance fluctuates significantly even for neighboring checkpoints. However, unlike prior work we use a *fixed* evaluation set of 50 episodes in simulation to choose the best checkpoint. This reduces the likelihood of choosing the checkpoint that was evaluated on favorable environments (i.e., rejection sampling of harder environment initialization).

For all experiments, our method uses an RNN (LSTM) for Dense Net (predicting the mode and the dense action), and uses a separate MLP with the same inputs for the Sparse Net (predicting sparse waypoints), as shown in Fig. 1.

The input embedding is then passed into SparseNet (MLP) which outputs the waypoint as a robot pose (position and quaternion). DenseNet can be any sequential model (RNN, Transformer, etc) that produces some temporal embedding $\tau_e$ (RNN in our case). This architecture is shown in Fig. 6, and the training cycle is shown in Algorithm 2.

### C.3  Evaluation Details

During evaluation (see Algorithm 3), the policy chooses the mode using $\tilde{m}_t$. If $\tilde{m}_t = 0$, the model will servo in a closed-loop fashion to the predicted waypoint $\tilde{w}_t$ (Line 7) using controller T (Line 10). The policy is queried at every step to continually update the policy hidden state, but importantly its outputs are ignored until we reach the waypoint to avoid action prediction errors. (Line 4). If $\tilde{m}_t = 1$, the model will execute one step using the predicted dense action $\tilde{a}_t$ (Line 14).

### C.4  HYDRA Hyperparameters

The hyperparameters used in the main text for all six environments are shown in Table 1, for BC, BC-RNN, and HYDRA. Hyperparameters stay mostly constant for HYDRA across all of the experiments, with larger policy sizes for harder tasks. Additionally, in almost all cases, BC-RNN, BC, and HYDRA share the same hyperparameters where possible. In the real world experiments, hyperparameters are exactly the same both across methods and across environments.

We chose parameters through extensive parameter sweeps (for each column in Table 1 except the number of demos and batch size) and picking the best performing model for all methods. For

**Algorithm 3** Test Time Execution
___
1: Given env, $\pi(m, a, w|o)$, initial state $o_0$, controller T
2: $t = 0$, $w = $ None
3: **while** not done **do**
4:    $\tilde{m}_t, \tilde{a}_t, \tilde{w}_t \sim \pi(\cdot|o_t)$                                     ▷ Sample policy
5:    *// Check for new sparse mode*
6:    **if** $w$ is not set and $\tilde{m}_t = 0$ **then**
7:       $w = \tilde{w}_t$                                            ▷ Set a new waypoint
8:    *// Compute the waypoint-optimal action (sparse)*
9:    **if** $w$ is set but not reached and not timed-out **then**
10:       $\tilde{a}_t \leftarrow \mathrm{T}(o_t, w)$                          ▷ Compute waypoint-optimal action
11:    **else**
12:       $w = $ None                                 ▷ Unset waypoint if reached
13:    *// Step the environment*
14:    $o_{t+1} = $ env.step($\tilde{a}_t$)
15:    $t = t + 1$
___

*ToolHang*, we found BC-RNN with a horizon of 20 was overfitting due to the high dimensional input space, while HYDRA seemed more robust to that, potentially since more consistent actions leads to a simpler objective (see Section 4.1). For BC, using an MLP size of 1000 for *ToolHang* also seems to overfit and perform worse than an MLP size of 400, while the reverse was true for BC-RNN and HYDRA. We find that all methods are sensitive to these hyperparameters, which is an open problem for the community.

## D   Additional Results & Analysis

In this section we show rollouts of our method and baselines, and then perform ablations of our method and analyze the results, including mode labeling sensitivity, mode label learning from less data, choices in action space design, different loss weightings, and robustness experiments. All ablations are performed on the *NutAssemblySquare* task unless otherwise stated.

### D.1   Rollouts for Real Environments

Fig. 7 shows example rollouts from the uncurated demonstration, the learned BC-RNN policy, and HYDRA. Qualitatively, in the top row of Fig. 7 we see that HYDRA produces more consistent and optimal trajectories at evaluation time that help the policy to stay within the narrow "band" of the successful state distribution at test time, thus improving performance.

For the long horizon *MakeToast* task, the performance of HYDRA is much better than BC-RNN, but lower overall than in *MakeCoffee*. We hypothesize that the difference between this task and *MakeCoffee* is primarily in the consistency of demonstrated actions (see demonstration rollouts in Fig. 7), with significant variation in the behaviors for nearby states especially during dense periods. This leads to BC-RNN having highly noisy and sub-optimal actions, which manifest quite noticeably in Fig. 7. However, HYDRA yields much more consistent and optimal motions, reducing the distribution-shift problem.

### D.2   Mode Sensitivity

Next, we consider the sensitivity of HYDRA to mode labels, specifically in terms of the number of labeled waypoints in each episode. In Table 2, we ablate the number of waypoints by introducing N intermediate waypoints in every sparse segment, for $N = 1$ and $N = 2$. Since there are at least 3 sparse segments labeled in each demo in *NutAssemblySquare*, this corresponds to adding at least 3 or 6 more waypoints to each demonstration, respectively. We see that performance drops are relatively minor in both cases, showing that HYDRA is robust to different waypoint choices. We hypothesize that the reason for the minor performance change when adding more waypoints is that SparseNet must learn a more complex waypoint space that is more multi-modal.

To understand the sensitivity to *consistency* of mode labels, we also show the results for an aggregated dataset consisting of two mode labelers in Table 2. We find that using multiple labelers does not result in a huge drop in performance, however there is still some sensitivity to having different mode labeling strategies amongst the two labelers.

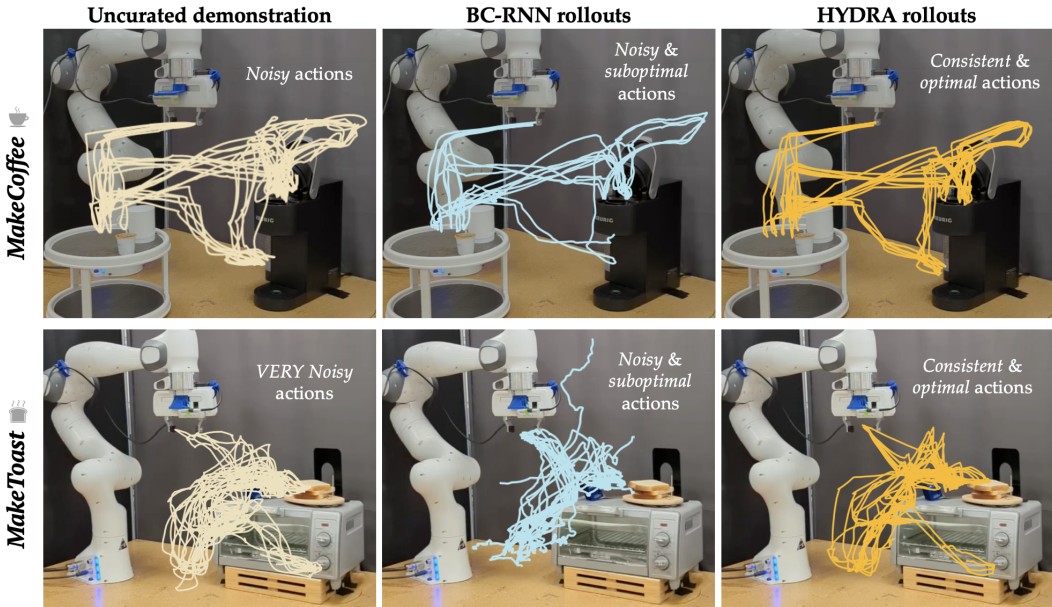

| | Uncurated demonstration | BC-RNN rollouts | HYDRA rollouts |
|---|---|---|---|
| MakeCoffee | Noisy actions | Noisy & suboptimal actions | Consistent & optimal actions |
| MakeToast | VERY Noisy actions | Noisy & suboptimal actions | Consistent & optimal actions |

Figure 7: *MakeCoffee* (top) and *MakeToast* (bottom) rollouts, with the demos (left), HYDRA rollouts (middle), and BC-RNN rollouts (right). Our method produces more consistent and optimal actions compared to both BC-RNN and the demonstrations, and thus is able to stay within the narrow success "band" of the state distribution. BC-RNN has many sub-optimal behaviors, leading to less completed trajectories in middle column. The demonstrations for *MakeToast* are even noisier than those in *MakeCoffee*, leading to even more noticeable distribution shift for BC-RNN in the *MakeToast* task. In contrast, HYDRA *curates* the demonstrations in *MakeToast* using sparse and dense periods to follow more consistent paths, thus leading to higher success.

| Base | Add-1 | Add-2 | Two-Labelers |
|---|---|---|---|
| 90.0 | 86.0 | 80.0 | 86.0 |

Table 2: Success rates for HYDRA when artificially more waypoints are added to sparse periods (left) and for an aggregated dataset with two labelers (right). Adding intermediate waypoints to sparse segments has only a minor effect on performance despite the increase in complexity of the pose action space. Likewise using multiple labelers has a minor drop in performance.

## D.3   Learning Mode Labels from Less Data

Providing mode labels can be an additional overhead when training HYDRA. To reduce overhead, we might want to learn the mode labels from a few labeled examples, and use this to relabel the rest of the dataset. To show the promise of such an approach, we learn to predict the "click state" at each time step (same as in Fig. 5) using a simple RNN architecture with the same parameters as the model used for training. This model outputs two logits, one for the mode itself ($m_t$), and one that represents a switching criteria between segments ($s_t$). This allows us to predict not only the sparse or dense label, but also the waypoint label for each sparse segment. We additionally smooth both $m_t$ and $s_t$ as is commonly done in binary sequence prediction tasks. In Table 3, we demonstrate that we can learn mode labels from 25% of the data with only a 10% drop in performance for the square task, and even less of a drop when training on 50% or 75% of the data.

| 90% | 75% | 50% | 25% |
|---|---|---|---|
| 92 | 88 | 86 | 82 |

Table 3: Success rates for HYDRA for *NutAssemblySquare* when the mode labels are learned (predicting "click state" in Fig. 5).

With this preliminary evidence, we believe the sample efficiency of this mode learning procedure can be improved by incorporating prior data from a wide range of tasks, potentially even using labeled internet data. To address the multi-modality of mode labels that might occur when having multiple

people provide labels, future work might leverage few-shot or in-context learning approaches to adapt to a particular *style* of mode labeling.

## D.4 Variations in the Action Space

Why do we need the dense period at all? In Table 4, we compare HYDRA's hybrid action space to waypoint only ablations, both with and without the test-time controller $T_{linear}$. With $T_{linear}$, the model outputs a waypoint and the robot reaches that waypoint using $T_{linear}$ without querying the policy ("open loop"), and without $T_{linear}$, the model outputs a new waypoint every step which gets converted to action $a$ using $T_{linear}$ ("close loop").

First we show results for WP-Next{N} in Table 4, where waypoints are the pose of the robot N steps in the future at each state (hindsight relabeling). Second, we compare to WP-Mode, which uses the same mode labels in HYDRA to get more intelligent future waypoints during sparse segments. No pose-based models see any success, which we hypothesize is due to the mismatch between the human action $a$ and the online action $T_{linear}(o, w)$, which can lead to out of distribution states. Even in the open loop case, the waypoint only models are unable to perform the task, with failures involving imprecise behaviors during dense periods where exact velocities truly matter.

We additionally compare our method with and without the use of $T_{linear}$ online (first row in Table 4). We see that HYDRA greatly benefits from the online waypoint controller, since $T_{linear}$ follows an optimal path while the policy-in-the-loop approach leaves room for compounding errors in both the mode, action, and waypoint prediction. This once again illustrates that HYDRA yields more consistent and optimal actions by employing a hybrid action abstraction.

|  | Ours | WP-Next1 | WP-Next2 | WP-Next5 | WP-Mode |
|---|---|---|---|---|---|
| w/ T | 90.0 | 0.0 | 0.0 | 2.0 | 0.0 |
| w/o T | 58.0 | 0.0 | 0.0 | 0.0 | 0.0 |

Table 4: Success rates for different action spaces. HYDRA uses a hybrid action space, while the the rest use a pose-based action space. Top row: waypoints are reached using $T_{linear}$ before calling the policy again ("open" loop). Bottom row: waypoint actions are computed at every step and instead of reaching the action, the policy will convert a waypoint $w$ to dense action $a$ using $T_{linear}$ ("closed" loop). WP-Next{N} uses the proprioceptive state N steps in the future as the waypoint for each state. WP-Mode uses the same mode labels as in HYDRA to get the waypoints, but does not implement a hybrid action space. None of the pose-based action spaces get reasonable performance, showing the importance of both dense actions and waypoint phases.

## D.5 Ablating Mode Weighting ($\gamma$)

We also show the effect of different values of $\gamma$, the weight of the current mode loss. If for a given step in training mode $m_t = 0$ (sparse), then we weight the sparse waypoint loss for $w_t$ with $1-\gamma$ and the dense action loss for $a_t$ with $\gamma$. Lower $\gamma$ thus corresponds to fitting the current mode action loss more than the other mode's loss. Therefore, $\gamma$ also controls the contribution of the relabeled actions during sparse periods to the overall objective in Eq. (2). We use $\gamma = 0.5$ in most experiments, meaning both action (waypoint and dense action) losses are weighted equally during training. We provide a sweep over $\gamma$ in Table 5 for *NutAssemblySquare* and *ToolHang*, and we see that choosing $\gamma$ only has a minor effect. Nonetheless, $\gamma = 0.5$ is consistently the best. This illustrates that (1) HYDRA is fairly robust to $\gamma$, (2) learning relabeled dense actions during sparse periods and sparse actions during dense periods is beneficial to performance – this supports the claim in Section 4.1 that training on relabeled dense actions outperforms uncurated dense actions.

|  | $\gamma = 0.1$ | $\gamma = 0.2$ | $\gamma = 0.4$ | $\gamma = 0.5$ |
|---|---|---|---|---|
| Square | 80.0 | 84.0 | 88.0 | 90.0 |
| ToolHang | 60.0 | 62.0 | 58.0 | 64.0 |

Table 5: Success rates for different values of $\gamma$ for both *NutAssemblySquare* and *ToolHang*. For both *NutAssemblySquare* amd *ToolHang*, $\gamma$ does not have a large effect. We saw even less of a change for vision based experiments. Thus for real world experiments, we fix $\gamma = 0.5$ (no mode-specific weighting).

One issue with our framework is the impact of false negative mode predictions, due to the use of the waypoint controller online; however, we mitigate the effect of this by training SparseNet even during dense periods (waypoint is the next state). Thus HYDRA is often able to overcome false negatives and still continue on the correct trajectory, providing another chance at the next step to

predict the correct mode label. On first glance, it is strange that $\gamma = 0.5$ consistently works the best in practice. We speculate that the choice in gamma actually helps control the issue with false negatives. Specifically, $\gamma = 0.5$ means that SparseNet will be trained equally during dense periods, and vice versa for DenseNet. We believe this helps overcome challenges of both false negatives or false positives.

## D.6 Transformer-based architecture

In Table 6 we show the performance of a purely transformer-based BC implementation on the *KitchenEnv* task. We see in this long horizon task that BC-RNN notably outperforms BC-Transformer in this single-task imitation learning setting, and we found similar drops in performance for the state-based simulation experiments. Thus, we did not include BC-Transformer as a baseline in our real world experiments. We note that VIOLA, which uses a similar underlying transformer but with a object-centric input representation, performs notably better on *KitchenEnv* than BC-Transformer.

|  | BC-RNN | BC-Transformer | VIOLA | HYDRA |
|---|---|---|---|---|
| Square | 84 | 78.0 | – | 90.0 |
| Kitchen | 52.0 | 24.0 | 78.0 | 87.0 |

Table 6: Success rates for different values of BC architectures on *NutAssemblySquare* (state-based) and *KitchenEnv* (vision-based). For *NutAssemblySquare*, we see that using BC-Transformer minorly reduces performance. In *KitchenEnv*, we see a larger performance drop for BC-Transformer compared to BC-RNN. VIOLA proves a superior transformer based architecture compared to simple BC-Transformer for *KitchenEnv*. In all cases, HYDRA beats both RNN and Transformer-based baselines. All models share the same visual encoder structure and action spaces as described in Table 1.

## D.7 Robustness of HYDRA to system noise

In Section 3 we noted the fundamental trade-off between consistent actions and state diversity. HYDRA breaks this tradeoff by relabeling actions in offline data, encouraging action consistency without reducing the state coverage of the data. To show that HYDRA still benefits from the state diversity in human data, in Table 7 we analyze the effect of system noise on HYDRA and BC. We find that HYDRA only drops from 90% to 86% (4% drop) under the same system noise as used with BC. This shows that not only does HYDRA capture the state diversity in human data, but it is able to be even more robust to distribution shift than BC. We attribute this boost in part to the use of a closed loop waypoint controller, which consistently reaches the waypoint under system noise. This also supports the claim made in Section 6 that the gap in performance between HYDRA and baselines in real compared to simulation experiments can in part be attributed to the added system noise found in the real world.

|  | Base | Noise=0.1 | Noise=0.3 |
|---|---|---|---|
| BC-RNN | 84 | 76.0 | 60.0 |
| HYDRA | 90 | 92.0 | 86.0 |

Table 7: The effect of increasing system noise (columns left to right) on BC-RNN (top row) and HYDRA (bottom row) trained on human data for *NutAssemblySquare*. While BC-RNN drops 24% under the max system noise, HYDRA only drops 4%, illustrating the ability of HYDRA to capture state diversity and thus be robust to distribution shift.

## D.8 HYDRA-NR Analysis

In Section 5 we see that HYDRA without action relabeling (HYDRA-NR) often performs worse than HYDRA, due to less action consistency in the dataset. However, we see in Fig. 3 that for the 50 episode domain, performance is actually similar for HYDRA and HYDRA-NR. We suspect this is because there is so little state coverage that the action variability is artificially low, and thus the benefit to relabeling is minor. In the kitchen environment, we also do not see much change for HYDRA-NR, which we speculate is due to higher action consistency in the demonstrations in the dataset. Qualitatively we observed that the kitchen environment demonstrations are much higher quality than other environments, and also involve a much smaller action space (only position control, no orientation, consistent with prior work).

