# OpenReview forum: "HYDRA: Hybrid Robot Actions for Imitation Learning"
_robot-learning.org/CoRL/2023/Conference — CoRL 2023 Poster_

### Official Review · Reviewer_gKhQ · 2023-06-27

**Confidence:** 4
**Originality:** Very Good
**Technical Quality:** Good
**Clarity Of Presentation:** Excellent
**Impact:** 3

**Recommendation:**

Weak Accept: I recommend accepting the paper, but will not argue for my recommendation if the majority of other reviewers have a different opinion.

**Review:**

## Strengths
- The paper is overall well-organized and well-written. I find the paper enjoyable to read. The motivation of the paper is clear. The equations and notations within the paper are well-explained. The figures are well-made and nicely explains the main idea of the paper.
- I find the proposed method novel (to my knowledge) and interesting. The idea of using a hybrid action representation is quite neat. The usage waypoint reduces the effective horizon of the task, while being able to predict per-step action greatly improves the flexibility of the proposed approach.
- The experimental evaluation is on a real robot with visual observation. The tasks considered are non-trivial, some involve long horizons.
- The limitation discussion is thorough. I appreciate that the authors also show preliminary experimental results on potential future work.

## Weaknesses
- Sometimes, it is unclear why the authors do not include certain baseline algorithms or ablation studies. For example, why is HYDRA-NR not provided for “NutAssemblySquare” with 100 and 50 demonstrations and “KitchenEnv”? Why is BC not included for KitchenEnv? Why is BC not included for all real robot experiments? Why do the authors only present VIOLA results for “KitchenEnv” and “MakeCoffee”, but not for others? I can see the experimental results much more convincing if all these baseline results are presented in the paper, even if HYDRA does not outperform these baselines.
- The proposed method requires the labeler to identify the mode. Similar to the fact that BC might not work well with inconsistent actions, it seems that the proposed method may not work well when the mode annotations are not consistent. This could happen especially when the modes are given by different labelers. In Appendix A, the authors mention that HYDRA is robust to “the labeling strategies across the two labelers”. It is unclear if it refers to training HYDRA with each labeling strategy separately or training HYDRA with an aggregated dataset by two labelers.
- The demonstrations for the real robot experiments are given by a “proficient user”. It would be interesting to see if HYDRA is also effective given demonstrations (including but not limited to mode labeling) from inexperienced users.
- For long-term tasks, sometimes there is not an absolute ordering for each step. For example, for the “MakeCoffee” task, the robot can achieve the task by either first inserting the lid then placing the mug, as described in the paper, or the other way around. It seems that the demonstrations are intentionally given in a consistent order. It would be interesting to see if HYDRA can also work with demonstrations with inconsistent ordering of intermediate steps.

## Post Rebuttal Updates
I believe most of my concerns have been adequately addressed during the rebuttal period. As I noted in my original review, I find the paper enjoyable to read and the proposed approach interesting. I also find the results to be more convincing with the additional numbers from HYDRA-NR. I am still not very convinced that comparison with VIOLA is not needed due to VIOLA's complexity. Nonetheless, I find the paper worth publishing.

**Quality Of The Limitations Section:**

Limitations are addressed clearly

**Questions For Rebuttal:**

- Why are some baselines/ablation results missing from certain experiments (see the first point in **Weaknesses**)?
- Regarding hyperparameters (Table 1 in Appendices), why does HYDRA use a different horizon (20) than other algorithms for “ToolHang”? - - Why does BC use a much smaller hidden size (400) than other algorithms for “ToolHang”? How are those hyperparameters determined?
- In the introduction (line 40), can the authors provide an intuitive explanation on why a waypoint-only approach could fail to “insert a coffee pod”?
- Why not use a recurrent architecture for the sparse net as well?
- When $m_t=0$, the waypoint controller is used until either the waypoint is reached or a timeout. This strategy seems to be vulnerable to false negatives of mode labeling. What is the consideration behind not just letting the waypoint controller to execute for one step then re-evaluate?
- Although the authors provide a discussion on the choice of $\gamma$ in Appendix D.5. I still find it puzzling that $\gamma=0.5$ (not differentiating between modes) performs the best consistently. Can the authors provide a hypothesis on why this is the case? Is it related to test-time errors of the mode predictor?

**Robotics Focus:**

Sufficient demonstration on hardware

**Summary Of Paper:**

The paper proposes a hybrid action representation for imitation learning. The policy outputs both 1) the action (which is the same as standard behavior cloning) and 2) a waypoint which is given to a lower-level waypoint controller. The policy also produces a “mode”, which determines which of the two representations is used. The mode within demonstrations can be labeled both in an online and offline fashion. In the waypoint mode, the actions in the  demonstrations are relabeled by the waypoint controller. The waypoint mode helps reduce effective horizon and improve data consistency. The simulated and real-robot experiments show that the proposed strategy is effective in learning long-horizon tasks compared with existing baselines.

**Summary Of Recommendation:**

The paper is well written and technically sound. I like the proposed approach and find it, to my knowledge, novel. My major concerns lie in the experimental results of this paper. I find the evaluation to be not very convincing due to missing or rather, inconsistent choices of baselines. I will consider increasing my score if the authors can either justify their choices of baselines or provide the results for the missing baselines.

---

### Official Review · Reviewer_1bNM · 2023-07-19

**Confidence:** 4
**Originality:** Good
**Technical Quality:** Good
**Clarity Of Presentation:** Very Good
**Impact:** 3

**Recommendation:**

Weak Accept: I recommend accepting the paper, but will not argue for my recommendation if the majority of other reviewers have a different opinion.

**Review:**

The paper proposes a nice trade-off between learning from predefined temporal abstractions and learning only from low-level expert actions. The problem is clearly explained, along with the relevant related works. The method is quite simple, yet very well described and evaluated. The results seem promising, especially in the real-world domains tested.
The main advantages of the method are that it can maintain dexterity of the policy when it is required by the task and, at the same time, avoid cumulating errors in the state during sparse interactions with the environment.
The main limitations of the approach regard the amount of additional labels and prior knowledge that it requires. In fact, the method needs both: (i) some extra labels specifying the mode to use at each time step, which must be collected from the experts, and (ii) a controller T to use during preprocessing of the dataset for relabeling the low-level actions and to be used at test time. Regarding the controller, the authors say: "the main requirement for labeling modes is that during sparse phases (mt = 0), the labeled waypoint wt should be reachable via T starting from ot (i.e., without collision)". They say that this reachability requirement can be ignored in case a collision-avoidance planner is available, but, however, they do not assume to have such a planner, and they do not provide any explanation on how they manage possible collisions while using the controller. I want the authors to better elaborate on this point, since it is a crucial feature of the approach that could eventually compromise the framework's applicability. This is one reason why I propose only weak acceptance for the paper. Another one regards the comparison with other methods. They only compare their approach with behavior cloning (BC) and BC-RNN. I wonder how their method could compare with other imitation learning algorithms and with Johns 2021, on which they build their approach.
Overall I found the paper clear, except for the part I underlined, and the approach quite novel and fairly evaluated.

**Quality Of The Limitations Section:**

Limitations are addressed clearly

**Questions For Rebuttal:**

How do you avoid possible collisions during the use of controller T?

**Robotics Focus:**

Sufficient demonstration on hardware

**Summary Of Paper:**

The paper proposes HYDRA: a hybrid method for robot learning from expert demonstrations. Policies learned through Imitation Learning can suffer from state distribution shift at test time due to compounding errors in action prediction which lead to previously unseen states. The paper contributes to mitigating this issue by defining a two-level action representation: sparse high-level waypoints and dense low-level actions. HYDRA can switch between the two modalities online, leveraging high-level actions during free-space motions and dense low-level actions during contact-rich manipulations, so to maintain policy dexterity and generality at the same time. The framework is a multi-head network that requires additional mode labels to be annotated by the experts during data collection or offline. The authors evaluate their proposal in many simulated and real robotic tasks showing competitive results.

**Summary Of Recommendation:**

In summary, I recommend accepting the paper, because it is clear and proposes an interesting approach for combining high-level and low-level actions in imitation learning for robot tasks. The evaluation is not very extended, but still promising. The benefits and the limitations are overall well described, but some clarifications on the use of the controller in case of obstacles are needed.

---

> ### Author Response · Authors · 2023-08-14
> **Rebuttal Period Closing**
>
> We'd like to remind the reviewer that the rebuttal period is ending. We truly appreciate your valuable feedback on our paper. As the rebuttal period concludes tomorrow, we assure you that we have included all the discussion points brought up in the review in our revision and hope we have addressed your concerns. If there are any additional points you'd like us to discuss, please let us know!

---

### Official Review · Reviewer_QMAq · 2023-07-19

**Confidence:** 4
**Originality:** Good
**Technical Quality:** Good
**Clarity Of Presentation:** Good
**Impact:** 3

**Recommendation:**

Weak Reject: I recommend rejecting the paper, but will not argue for my recommendation if the majority of other reviewers have a different opinion.

**Review:**

Remarks and Questions:

- It seems like the data labeling method performs linear interpolation between cartesian end-effector locations:
- The authors describe this low-level action relabeling as the application of a “linear controller” to “imagine” new actions. This description is unnecessary, and also confusing (the use of “imagining” has typically been used for generative, learning-based methods which predict high-dimensional observations).
- How are end-effector poses interpolated during the re-labelling process? This would seem important for learning to align the gripper in the right way when approaching a grasp point.
- There seems to be an assumption in the linear-interpolation scheme that the linear trajectories are feasible, or can be executed by an underlying controller. If it is the latter,  please describe the controller used. Otherwise, please state how the approach deals with kinematically-feasible locations, singularities, etc.
- There also seems to be an assumption that the action space must be directly visible by the human annotator for relabelling. How would this approach be applied for cluttered/partially observable workspaces? How about higher-dimensional action spaces (ex. joint-space) that may be required for more complex tasks or constraint satisfaction (ex. obstacle collision avoidance).
- Sparse data labeling for robot Learning-from-Demonstration is an area that has been studied for a long time in Human-Robot-Interaction community, ex. through the use of key-frame demonstrations (see work done by Andrea Thomaz, for instance). There can be much variability in providing such coarse/sparse demonstrations, and user studies are generally required. How does this work relate to this body of work? How do they deal with consistency of trajectory segmentation across human demonstrators?

Pros:
- Generally well written, with clear presentation and figures.
- Evaluations are performed on a wide array of tasks on a physical robot system.

Cons:
- Baseline setup is unclear: how were BC-RNN, BC trained and on what data? Was this the same sparse/dense data as HYDRA? It would be more convincing to compare against BC-RNN/BC trained on only dense data from both modes (i.e. no action-relabeling).
- Unclear whether action relabeling has a large impact, judging by the experimental results. It may be beneficial to include the HYDRA-NR evaluations in more of the environments.
- IL baselines: given that mode-labeling can be performed online, it would be useful to compare against intervention-based methods, or expert-relabeling (ex. Dagger), at least on a subset of experiments.

**Quality Of The Limitations Section:**

Additional details required

**Questions For Rebuttal:**

Please address the points mentioned above.

**Robotics Focus:**

Sufficient demonstration on hardware

**Summary Of Paper:**

The paper addresses the issue of distribution shift in imitation learning for robot motion generation. They argue the importance for appropriate selection of action representation in combating this effect. Specifically, they propose to use a policy architecture designed with a hybrid action representation, combining sparse waypoint prediction for movement in free-space with dense action prediction for movement near and at contact/interaction points. The network also learns to predict switching between these two modes. The authors also argue for the importance of relabeling actions for improved temporal consistency and waypoint following.

**Summary Of Recommendation:**

The work is extensive, but the originality and novelty of the approach is unclear. More evaluations are needed to support the claims in the paper.

---

> ### Author Response · Authors · 2023-08-14
> **Rebuttal Period Closing: have we addressed your concerns?**
>
> We'd like to remind the reviewer that the rebuttal period is ending. We truly appreciate your valuable feedback on our paper.
>
> As the rebuttal period concludes tomorrow, we hope that we have clarified any confusion and addressed your concerns in our response. If there are any additional points you'd like us to discuss, please let us know.
>
> We kindly request your consideration for a potential score adjustment based on our responses. We thank the reviewer very much for your time reviewing our work!

---

### Official Review · Reviewer_DQGz · 2023-07-24

**Confidence:** 4
**Originality:** Very Good
**Technical Quality:** Very Good
**Clarity Of Presentation:** Excellent
**Impact:** 4

**Recommendation:**

Strong Accept: I recommend accepting the paper and will argue for my recommendation even if other reviewers hold a different opinion.

**Review:**

Quality: This paper addresses an important challenge in imitation learning (reducing compounding errors for solving long-horizon tasks,) is theoretically sound, and well-placed in the existing literature.

Clarity: The paper is well-written and easy to follow. The experimental setups are verbose enough to make them fairly reproducible, with additional details in the appendix.

Strengths:
1. The authors were able to clearly lay out the different classes of approaches that can be beneficial for mitigating compounding errors in imitation learning (introducing temporal abstractions in the policy and increasing model expressivity) and place their approach well in the existing literature.
2. The experimental results show that the proposed method maintains a much higher success rate for the most part of the multi-step tasks compared to baselines. The authors also identify the potential benefits of their method when working with real-world demos with higher variance.

Weaknesses:
1. For the MakeToast and SortDishes tasks, the proposed method did not perform well on the last few steps of the task. It would have been nice if the authors provided some compelling reasons for the failure.


**Quality Of The Limitations Section:**

Limitations are addressed clearly

**Questions For Rebuttal:**

1. Did the authors do any ablation or analysis to prove that the 2-level action space reduces compounding errors due to the state distribution shift when deploying this policy at test time?
2. The authors argue that making the action space more expressive to capture human multi-modality, coupled with consistent action demonstrations from the expert make the policy learning easier. Wouldn’t reducing diversity in the training data hamper potential usability of the policy when deployed in the real world? If so, can the authors identify tasks where this methodology would hamper performance?
3. It is unclear why VIOLA was omitted for the MakeToast and SortDishes tasks for experimentation. Can the authors clarify?


**Robotics Focus:**

Sufficient demonstration on hardware

**Summary Of Paper:**

This paper addresses the challenge of compounding errors in action prediction for long-horizon tasks when deploying imitation learning policies during test time. They introduce a 2-level policy that outputs sparse high-level waypoints at the higher level and dense low-level actions at the lower level. They argue that the temporal abstraction in the action space reduces the number of steps in the action space for the same horizon length thus reducing compounding errors due to the state-distribution shift during evaluation. Additionally, they argue that relabeling the demonstration dataset to contain consistent actions for the same state make learning dense actions easier. They show experiments in both simulation and real robots to prove that their method beats the baseline methods on long-horizon tasks, and provide explanation why their methodology helps when working with high-variance real-robot demonstrations.

**Summary Of Recommendation:**

The authors provide novel ideas for mitigating compounding errors when rolling out imitation learning policies, and show compelling evidence in the form of success rates on long-horizon tasks. They were able to articulate the proposed methodology well, making the paper theoretically sound and easy to follow. The contributions are robotics focused and open up some interesting venues for future research towards solving long-horizon tasks.

---

> ### Author Response · Authors · 2023-08-14
> **Rebuttal Period Closing**
>
> We'd like to remind the reviewer that the rebuttal period is ending. We thank the reviewer again for your positive comments and we truly appreciate your feedback on our paper. As the rebuttal period concludes tomorrow, we assure you that we have included all the discussion points brought up in the review in our revision and hope we have addressed your concerns. If there are any additional points you'd like us to discuss, please let us know.
>
> We thank the reviewer again for your time reviewing our paper!

---

### Author Response · Authors · 2023-08-12
**Shared Response to All Reviewers**

We thank all the reviewers for their detailed and constructive feedback. We are excited that the majority of reviewers agree that HYDRA is novel, evaluated extensively, and promising overall for long horizon IL.

In this shared response, we outline the points of feedback that came up several times and/or we thought were very important. A revised PDF is attached to each rebuttal response (changes in red).

**Baseline Choices**

Several reviewers wondered about our choice in baselines. Some specific requests were for HYDRA-NR to be evaluated in more settings, which we have now added (see response below), and for justification of why BC and VIOLA were not included in all experiments. We agree that providing a fair and comprehensive set of baselines is very important to evaluate HYDRA adequately. For example, BC was not included in Kitchen or the real world tasks since it was substantially lower performing than BC-RNN, both in our state-based simulated results and in some initial testing on visual domains. For VIOLA, as noted in the text, BC-RNN was shown to be a strong baseline in the MakeCoffee task, nearly matching VIOLA’s performance despite using a much smaller network architecture. Secondly, VIOLA takes longer to both train and perform inference due to the use of a region proposal network and a large transformer architecture. Thirdly, VIOLA changes the *input* and encoding structure on top of an approach like BC-RNN, whereas HYDRA changes the *output*. Thus these methods are actually compatible with each other, and the point of including VIOLA was not as a primary baseline but to show that designing intelligent *output action* spaces can be as or more beneficial than large changes in the *input features* or encoding structure. This is on top of the fact that HYDRA also uses substantially fewer parameters than VIOLA, and thus BC-RNN is a more directly comparable approach. Given these limitations, and our strong BC-RNN baselines, we argue the addition of VIOLA does not provide any essential signal about HYDRA. We have added this discussion to Appendix A.

**HYDRA-NR in more settings**

We recognize the need for more evaluation of HYDRA-NR in more settings to justify our claims about action relabeling. Thus we include HYDRA-NR results for both 100 episodes and 50 episodes in Square, as well as in the kitchen task (see revised Figure 3). We find that often, HYDRA-NR matches or is lower in performance than HYDRA. A new finding is that for the 50 episode domain, performance is actually similar for HYDRA and HYDRA-NR. We suspect this is because there is so little state coverage that the action variability is artificially low, and thus the benefit to relabeling is minor. In the kitchen environment, we also do not see much change for HYDRA-NR, which we speculate is due to higher action consistency in the demonstrations in the dataset. Qualitatively we observed that the kitchen environment demonstrations (which we also did not collect) are much higher quality than other environments, and also involve a much smaller action space (only position control, no orientation, consistent with prior work). We now include this discussion in more detail in Appendix D.8

**Mode Labeling & Consistency**

Reviewers noted that HYDRA requires additional mode labeling to extract waypoints and dense periods. However, we emphasize that (1) as we show in our real world experiments, mode labels can be provided during data collection with little additional demonstration time or curation, (2) mode labeling is fairly robust to different labeling strategies as shown in Appendix D.2, and (3) modes can be learned from substantially less data as shown in Appendix D.3. Also, we are excited about the potential to learn an *adaptable* mode predictor from diverse multi-task data, through which we hypothesize the mode labeling process can be automated with just a few examples.

Reviewers also wondered about the required consistency of mode labeling strategies. This is definitely an important consideration, as we discuss in Appendix B.3. We found in practice that for a single user, keeping mode labels consistent was not challenging after a few practice demonstrations, however as noted in the text it would be interesting to do a full user study. Critically, we found keeping mode labels (a binary, high level signal) consistent is significantly easier than keeping actions (continuous, 7D signal) consistent at each state. Add to this the fact that mode labels can be learned from less data (Appendix D.3), and we argue that the mode consistency problem is easier to solve than the action consistency problem. We have also added an experiment showing that an aggregated dataset from two mode labelers does not substantially lower performance in Appendix D.2.

---

> ### Author Response · Authors · 2023-08-12
> **Full Changelist in Revised PDF (see Rebuttal attachment per review)**
>
> **Full Changelist**
>
> - Added HYDRA-NR in all simulation tasks (Figure 3)
> - Results for two mode labelers in Appendix D.2
> - Discussing last stage of MakeToast / SortDishes in Section 5
> - Controller details & Baseline Setup in Section 5
> - Rewording “online” and “offline” mode labeling to “during” and “after”
> - Added keyframe demo citations in related work
> - Justifying baseline choice in Appendix A
> - Collision-free control details in Appendix B.3
> - Hyperparameter sweep details in Appendix C
> - Discussing false negatives for mode prediction in Appendix D.5

---

### Decision · Program_Chairs · 2023-08-30

**Decision:**

Accept (Poster)

**Comment:**

This paper presents a new method for imitation learning, which predicts either coarse, sparse actions (for free space motion) or fine-grained, dense actions (for object interaction). This requires the human operator to specify, for each stage during the demonstration, if that stage is a “coarse” stage or a “fine” stage. The learned policy then predicts which of these two modes should be executed, as well as the action for that mode. Simulation and real-world experiments show that this method gives competitive results.

Reviews before the rebuttal were 1 x “weak reject”, 2 x “weak accept”, and 1 x “strong accept”. Authors addressed the reviewers’ comments, and provided an updated paper. This also included some new experiments that reviewers had asked for. Following the rebuttal, reviewers maintained their pre-rebuttal scores.

The AC and reviewers then had a brief discussion on the paper. Based on this, and the reviews, there is a general consensus that the proposed method is sensible and achieves good performance. Most of the issues raised by the reviewers were answered adequately by the authors and incorporated into the updated paper. The “weak reject” reviewer did not engage with the authors following the rebuttal, but they did not raise any major criticisms in their original review and mainly requested clarifications and new experiments (HYDRA-NR experiments, which the authors then provided). However, some reviewers found that the novelty is somewhat limited with respect to other methods (see below), and that the need for the operator to manually label different phases of the task might make this method impractical if non-experts wish to teach robots new tasks (a limitation noted by the authors). But overall, the reviewers and AC found the method to be sensible, effective, and interesting to the wider robot learning community.

An important point raised during the AC/reviewer discussion was that of novelty and the paper's related work section. Authors state that they build upon [36], but claim that [36] “cannot easily scale to multi-step manipulation tasks with multiple stages of unstructured object interaction.” However, in a CoRL 2021 paper “Learning Multi-Stage Tasks with One Demonstration via Self-Replay”, [36] was indeed extended for multi-step tasks, with a simple modification. Reviewers did not cite this work and so there is less novelty in HYDRA than is claimed by the authors. We encourage the authors to address this in the final paper, and more clearly describe the relationship between HYDRA, [36], and the CoRL 2021 paper.